# Effect of Teaching Methods on Motor Efficiency, Perceptions and Awareness in Children

**DOI:** 10.3390/ijerph191610287

**Published:** 2022-08-18

**Authors:** Gaetano Raiola, Tiziana D’Isanto, Felice Di Domenico, Francesca D’Elia

**Affiliations:** 1Department of Political and Social Studies, University of Salerno, 84084 Fisciano, Salerno, Italy; 2Department of Human, Philosophical and Education Sciences, University of Salerno, 84084 Fisciano, Salerno, Italy

**Keywords:** sedentariness, motor patterns, teaching methods, extracurricular activities, PA

## Abstract

Currently, physical inactivity and sedentariness in children are becoming increasingly common, resulting in children’s poor ability to perform basic motor patterns. It is important to find strategies that instructors can adopt to improve awareness of the importance of physical activity for health and wellness, as well as motor efficiency. Two teaching methods can be used: prescriptive teaching and heuristic learning. The aim of this study was to compare these two methods to determine which is the most suitable for developing motor efficiency. An additional aim was to verify the children’s level of enjoyment and self-efficacy through questions on perceptions and, subsequently, on awareness of the activity performed distinctly from perception. The sample consisted of 28 children randomly divided into two groups: HEUR-L, performing activities using heuristic learning, a basic method in ecological approach; and PRES-T, using prescriptive teaching, a basic cognitive method. A motor-efficiency test (TEM) and a survey were administered. A two-way ANOVA with repeated measures was used to test differences in motor efficiency. A Chi-square (χ^2^) test was used to compare differences between groups in terms of perceptions in enjoyment and self-efficacy and, on a second test, awareness of the activity performed. The results were statistically significant (*p* < 0.05). Both teaching methods improved motor efficiency, although HEUR-L did so to a greater extent. Differences in perception were found in terms of enjoyment and self-efficacy (*p* < 0.05), whereas there was no difference in terms of awareness (*p* > 0.05). Although both methodologies led to improvements in motor-pattern development, heuristic learning was found to be the most effective method to improve motor efficiency, relationships and self-efficacy.

## 1. Introduction

### 1.1. Dissemination of Inactivity and Sedentariness in Children

Currently, physical inactivity and sedentariness in children is becoming increasingly common, a situation worsened by COVID-19 [1,2,3]. The “OKkio alla Salute” surveillance system, which monitors children’s lifestyles in primary school, states that children are moving too little. In the Campania region (Italy), 44.2% of children are overweight [4]. Approximately 3 of 10 children (31.4%) practice at least 1 h of structured sports activity 2 days a week, while regarding the time spent on physical activity (PA) games, only 2 of 10 children (20.3%) practice at least 1 h of activity 2 days a week. These data show that children practice little PA. It is estimated that one in three children is physically inactive, both girls and boys. Barely more than 4 in 10 children reach the recommended PA level for their age [4]. Moreover, children do not know how to move in the manner of previous generations. Performing basic motor patterns such as jumping, throwing, running and grasping is difficult for children due to their lack of coordination, mobility, strength, balance and speed. About two out of three children do not know how to run, jump or perform a forward flip [5]. This is due to the replacement of outdoor games with electronic devices and the poor availability of green spaces, public sports facilities and transport [4]. Excessive sedentariness and low PA are two important risk factors leading to chronic and metabolic diseases, such as cardiovascular disease or diabetes. Being physically active from an early age is important for health, as evidence suggests that PA behaviors or device use are established in early childhood and continued over time [6,7,8]. Therefore, children should develop a healthy and active lifestyle from an early age. According to the World Health Organization [9], the recommended levels of PA for children and adolescents (5–17 years) include all activities such as games, structured exercise, physical education, sport, travel, performed in the family, school and community context. Together, they allow an average of 60 min of daily movement during the week, and not ‘at least 1 h a day’, as previously recommended. Moderate-to-vigorous PA and muscle-strengthening exercises are also recommended at least 3 times a week.

### 1.2. Motor Efficiency to Assess Basic Motor-Pattern Development

It is necessary for movement experts, such as sports kinesiologists, physical education teachers and instructors of various disciplines, to take note of this and develop strategies in order to introduce and motivate children to motor practice in the form of play, to develop and consolidate basic motor patterns. To monitor the development of basic motor patterns, resulting from the training of coordination and conditional motor skills, the choice of the most suitable instrument is a relevant factor. A test containing exercises such as running, jumping, quadrupedal gait and throwing suitable for children aged 6–17 years is the motor efficiency test [10]. This test assesses motor efficiency, understood as a measure of physical abilities associated with coordination aspects.

### 1.3. Teaching–Learning Methods to Improve Basic Motor Patterns

Between the ages of 6 and 11, the motor load is identified with the continuous variation of methods, content and tools to achieve the optimal conditions to prevent developmental regression and ensure a correct start to movement and sport. Therefore, a very important aspect concerns the teaching styles used in the motor-sports field [11]. Very often, instructors prefer to adopt a reproductive style, which means the reproduction of movement patterns already known, demonstrated by the instructor and imitated by the pupil. This style can be traced back to the cognitive approach and the prescriptive teaching method, based on structuring motor programs and optimising their parameterisation. The techniques used are exercises that can be simplified, segmented, varied, randomised, etc. [12]. The objective is to stabilise and perfect the motor program while minimising executive variability. This style is preferred for its easy application, greater group management in dynamic activities, and reduced time for programming activities [13,14]. However, a lack of adequate stimuli due to the mere repetition of movements can decrease children’s interest and participation [15,16], slowing the acquisition of mastery of basic motor patterns. A second style, which is less often used, is the productive style, which means the production of new movement patterns by the child from given stimuli. This style can be traced back to the ecological–dynamic approach and heuristic learning, according to which action is directly available to those who act in their environment. The motor–sense system possesses self-organising properties that do not require the use of a motor program [17]. The ecological–dynamic approach favors heuristic learning, which aims to stimulate the emergence of spontaneous solutions to motor problems by exploiting executive variability through educational practices, such as techniques borrowed from psychology (circle time, brainstorming, cooperative learning...), the alteration of the environment, and game rules [12]. 

### 1.4. The Importance of Perceptions and Awareness in PA

Higher-quality motor experiences stimulate greater commitment and adherence to an active lifestyle, promoting moral and social development, positive self-perception and affection [18]. Although teachers play a crucial role in promoting such activities [19], there has been little effort to understand the effects of different teaching methodologies on the development of children’s basic motor patterns and their perceptions about it. A high level of perception of physical competence contributes to increased PA in children and young people, as does satisfaction [20,21]. Furthermore, becoming aware of the benefits of PA enables children to learn the importance of movement in everyday life, encouraging them to adopt a healthy and active lifestyle.

### 1.5. Aim of the Study

To summarise, the problem is the prevalence of poor PA in children, resulting in their poor ability to perform basic motor patterns. It is important to develop strategies that the instructor can adopt to improve awareness of the importance of PA for health and wellness, over motor efficiency, through the two teaching methods previously addressed. Consequently, the aim of this study was to compare the effects of prescriptive teaching and heuristic learning, and to verify the most suitable to develop motor efficiency; additionally, to verify the level of enjoyment and self-efficacy through questions on perception and, subsequently, awareness of the activity performed distinctly from perception. Children’s perceptions can help the instructor monitor the proposed activities in order to make them more attractive and diversified, according to all participants’ needs. On the other hand, awareness allows children to be more attentive to activities by understanding the reasons for and importance of their experiences.

## 2. Materials and Methods

### 2.1. Design and Participants

The study design was a pretest-posttest experiment with two groups. The sample consisted of 28 children, aged between 6 and 8 years, selected by convenience sampling from the province of Salerno in Campania (Italy). Children aged over 6 years were taken into consideration because at this age, children begin to have a greater awareness of the activities they perform; one aim of the study was to make children aware of the importance of practicing PA consistently to counteract sedentariness and physical inactivity [22].

The subjects were randomly divided into two groups, HEUR-L and PRES-T, which performed the same activity protocol using two different teaching methods, administered 2 days a week for a total period of 1 month. Informed consent was obtained from the parents of the children. Data were treated anonymously.

The instructor involved in the study was a kinesiologist with a degree in sports sciences, who was trained in the use of the two approaches to teaching physical education.

### 2.2. Educational Intervention

Subjects, after an initial assessment, were randomly assigned into two groups: HEUR-L (heuristic learning) and PRES-T (prescriptive teaching).

HEUR-L (*n* = 14; age, 7 ± 0.87 years old; sex, 50% M and 50% F; height, 1.31 ± 0.07 cm; weight, 39.5 ± 10.77 kg), consisting of children attending extracurricular lessons, performed 2 h of PA per week for a month, using the ecological–dynamic approach.PRES-T (*n* = 14; age, 7 ± 0.87 years old; 50% M and 50% F; height, 1.31 ± 0.07 cm; weight, 38.4 ± 7.43 kg), consisting of children attending extracurricular lessons, performed 2 h of PA per week for a month, using the cognitive approach.

#### 2.2.1. HEUR-L Protocol

In heuristic learning, the instructor does not enter into the trainees’ decision-making process. He leaves it to them through trial and error to find the solution to each problem. This approach promotes self-learning. The protocol consisted of:Warm-up. This consisted of exercises performed in rotation by the children based on material provided by the instructor, such as video tutorials or illustrations.Central phase. This consisted of an initial brainstorming activity on the theme of the day, viewing of video tutorials prepared by the instructor, followed by an exchange of opinions, and performance of the activities through the methodology of cooperative learning or peer tutoring.Cool-down. This consisted of circle time, in which each child gave their impressions of the activity, followed by some exercises to relax the body.

#### 2.2.2. PRES-T Protocol

In prescriptive teaching, the instructor is at the center of the educational action and is responsible for demonstrating each exercise in order to allow the student to memorise it through imitation and repetition of the motor gesture. The protocol consisted of:Warm-up. This consisted of some exercises to increase heart rate and body temperature, demonstrated by the instructor and performed by the children.Central phase. This consisted of routes and circuits prepared and demonstrated first by the instructor and then imitated by the children. The techniques used were partial exercises, which were varied and randomised so as to perfect execution through repetition of the individual exercises.Cool-down. This consisted of exercises to relax the body, demonstrated by the instructor.

An example of the activities performed in both groups is shown in Table 1.

### 2.3. Motor Efficiency

To check the effects of the two teaching–learning methodologies, motor efficiency was measured before and after one month. The study used the Motor Efficiency Test, or “Test dell’efficienza motoria” (TEM), an Italian tool developed by Cirami and Bonavolontà [10]. The test consisted of a circuit performed in 2 min, involving 4 stations and 4 gaits:Running around 4 cones in the same direction (speed)Keeping balance on rise for 10″ (balance)Monopodal and bipodal jumps in hoops (rhythm, quickness, differentiation)Bipodal lateral jumps (10) across two obstacles (lower limb strength, s/t orientation)Quadrupedal gait (rhythm, coordination)Throwing 3 balls of different sizes and consistency in a hoop (coordination, differentiation)Running backwards (s/t orientation)Bouncing the ball in a hoop to center a target (coordination, s/t orientation)

Results derived from overall speed, accuracy and endurance. For each exercise performed well, 1 point was given, except for exercise No. 6, for which 1 point was awarded for each hoop. At the end of the test, each child was required to provide his perception of effort involved using the Borg scale from 1 to 10 (RPE Scale for Kids). The Borg scale value was subtracted from the total score in order to obtain the motor-efficiency index. 

### 2.4. Surveys

#### 2.4.1. Perceptions

To check whether there was a difference in perception between the two groups regarding the effectiveness of the activity performed, a survey was administered containing the following questions and their response options:(1)Did you enjoy performing these exercises? Yes, no.(2)At the end of these activities, do you feel more improved in making friends with your peers or doing the exercises? Friendship, exercises.(3)Do you think that performing exercises with some difficulties can help you become better? Yes, no.

#### 2.4.2. Awareness

To check whether perceived activities became conscious, a survey was developed containing the following questions and their response options:(1)Given the significant improvements in both groups, do you agree that these two types of intervention are effective and motivating in order to promote well-being and break down sedentary lifestyles? Yes, no.(2)Given the greater improvements in HEUR-L than in PRES-T, do you agree that the educational practices are more optimal for developing/consolidating basic motor patterns? Yes, no.

### 2.5. Statistical Analysis

After verifying the normality of the data with the Shapiro–Wilk test and the homogeneity of variances with the Levene test, an independent samples *t*-test was performed to assess the differences between the pre-intervention groups. A two-way repeated-measures analysis of variance (ANOVA) was used to test for differences in the changes induced by the two protocols in the children’s motor efficiency. The independent variables included a between (intervention) factor with two levels (HEUR-L and PRES-T), and a within (time = 1 month) factor with two levels (pre- and post-intervention). To examine the influence of the intervention on the development of our dependent variable, we used this ANOVA to test the null hypothesis of no different change over time between the groups (intervention × time interaction). To qualitatively interpret the magnitude of the differences, the effect sizes were classified as small (0.2–0.5), moderate (0.5–0.8) and large (>0.8). Finally, a Chi-square (χ^2^) test was performed to compare differences between groups in terms of perceptions of enjoyment and self-efficacy and, on a second test, awareness of the activity was determined to estimate the significance between them. Significance was set at *p* < 0.05. Data were analysed using SPSS (IBM SPSS Statistics for Windows, Version 26.0., Armonk, NY, USA).

## 3. Results

The Shapiro–Wilk normality test results showed that the motor-efficiency values agreed with a normal distribution, *p* > 0.05, as Levene test results for homogeneity of variances, *p* > 0.05. An independent samples *t*-test showed that there were no significant differences between the two groups on entry, t(26) = 0.245, *p* = 0.809.

We used a two-way ANOVA test to assess the effect of two teaching methods on motor efficiency in children. As depicted in Table 2, the two-way ANOVA results revealed that there were significant differences at *p* < 0.05 in motor efficiency according to group, time, and interaction.

Between subjects, the difference was statistically significant F(1, 26) = 3.952, *p* < 0.05, partial η^2^ = 0.132The main effect of time on the motor-efficiency score was statistically significant, using Greenhouse–Geisser correction F(1, 26) = 470.361, *p* < 0.000, partial η^2^ = 0.948Training programs × time-interaction effect was also significant, using Greenhouse–Geisser correction F(1, 26) = 15.215, *p* < 0.001, partial η^2^ = 0.369

A detailed description of differences in motor efficiency pre–post intervention is depicted in Figure 1. 

The Chi-square analysis (χ^2^) revealed two associations. The first concerned the differences in perception between the two groups and question no. 2 and question no. 3. There was no relationship between the two groups and question no. 1. A detailed description is shown in Table 3.

No relationship was found between the two groups and their responses on awareness *p* > 0.05. A detailed description is shown in Table 4. 

## 4. Discussion

The present study aimed to compare the two teaching methods, prescriptive teaching and heuristic learning, in order to verify which led improvements in the learning of the basic motor patterns present on the motor-efficiency test and to verify the differences between the two groups in terms of perception and awareness of the activities performed. 

### 4.1. Motor Efficiency

The results showed us that both teaching methods were effective at improving the children’s motor efficiency, although HEUR-L, which used a teaching method based on heuristic learning, had a greater improvement than PRES-T, which used a method based on prescriptive teaching. Research showed that engagement in motor and sports activities was influenced by enjoyment, engagement alternatives, personal investment and social constraints [23]. Heuristic learning seemed to fully consider all these variables through teaching strategies, including brainstorming, circle time, peer tutoring, and cooperative learning. These allowed the children to express themselves freely and self-manage from simple instructions. Cooperative learning is a strategy used to structure learning activities that motivate students to engage in the completion of a motor task through interaction with peers. Subjects are divided into small groups, and members work together to achieve a common goal. An important element for implementing this strategy is the presence of equal groups. Each member has a different characteristic that can be exploited for a common purpose and monitored by the instructor [24]. Circle time is another strategy to improve various aspects of children, including self-esteem and the ability to communicate emotions and concerns so that the instructor can better plan subsequent lessons based on the feedback received. Since prescriptive teaching is limited to the simple repetition of exercises individually or in pairs, explained and demonstrated by the teacher, it offers fewer benefits. Furthermore, less work is undertaken on the motivational and socially affective aspects, which can lead to boredom and drop-out of motor practice. An important factor in teaching PA is the motivation of children to practice [25], although this has not been directly assessed. To facilitate the involvement of children, their interest must be constantly captured through strategies, including the use of non-traditional teaching methods, such as heuristic learning [26]. This involves the use of videos via the interactive whiteboard, discussions, peer-education and the exchange of information, which enables children to self-learn. Consequently, self-learning generates satisfaction, self-esteem and self-efficacy [15] because the child is able to learn independently, in its own time, and with its own tools and methods. 

Another key factor concerns the instructor, who must be knowledgeable about teaching methods and enthusiastic in conveying their passion for the discipline. Enthusiastic instructors motivate, inspire and excite students by promoting their learning [27,28]. An instructor who is well versed in teaching methods is able to cope with the problems children may encounter while performing the exercises by adopting different strategies to improve the emotional response to the exercise, which affects motivation to practice [29].

### 4.2. Perception

In terms of perception, both groups felt that the proposed activities were fun, regardless of the teaching method used. This is because the children enjoyed themselves with little, simply playing with their bodies, exploring the various movements and analysing the sensations they aroused [30,31]. On the other hand, a difference was found in the perceived effects of the activity practiced. HEUR-L stated that they improved more at establishing relationships with their peers, while PRES-T stated that they felt more prepared to perform the exercises. According to Gehris et al. [19], success in movement tasks strengthens children’s self-confidence, just as movement experiences help children develop social skills. This must be undertaken by the instructor, who must organise the learning experiences in the best possible way [32], working not only on the motor aspect but also on the social aspect. Knowing how to move easily helps children acquire greater emotional security in everyday contexts, as does knowing how to interact with other people. This can promote inclusion [33]. 

### 4.3. Awareness

In terms of awareness, we can see that there was no difference in perception between the two groups regarding the activity practiced. The first question asked whether the children agreed that these two types of intervention were effective and stimulating in order to improve well-being and break down sedentary lifestyles [34]. Almost all the subjects responded positively. This implies that the children are aware of the beneficial effects of movement and motor activities on well-being and leading a healthy lifestyle. The second question asked for confirmation that educational practices are the optimal teaching tools for developing/consolidating basic motor schemes. Again, no difference was found between the two groups. Interestingly, PRES-T, which used another teaching method, namely prescriptive, prefers the activities proposed using the other method, heuristic learning, as there is more interaction between peers, and it leaves room for freedom to express oneself [15].

### 4.4. Limitation of the Study

The study has some limitations, including the sample size and the use of a motor test that has not yet been validated. Future studies could extend the research by including a larger sample and assessing the long-term effects of the two teaching–learning methods. Furthermore, in addition to the physical aspect, a questionnaire assessing the motivational aspect could be administered, since although motivation appears to have been the key to HEUR-L prevailing over PRES-T, it was not directly assessed. Finally, it would be interesting to assess the instructor’s perceptions in terms of the teaching method habitually adopted and the satisfaction, effort and preference of two methods used.

## 5. Conclusions

Both teaching–learning methods improved motor efficiency and, consequently, basic motor patterns in children. However, HEUR-L, who performed the proposed activities according through heuristic learning, had greater improvements and perceived themselves as having developed more social skills and the ability to perform increasingly complex exercises. PRES-T, on the other hand, felt that they improved more in motor skills than in social skills and did not feel fully prepared for more complex exercises. Both were aware of the positive effects of these two methodologies in promoting well-being and, in particular, of the effectiveness of heuristic learning in the development/consolidation of basic motor patterns. Stimulating children in physical-motor activities enables them to consolidate basic motor skills, especially through activities that favour heuristic learning. Children enabled to move with a variety of group stimuli are more likely to move than to lead sedentary lives in the company of smartphones, video games, tablets and other technological devices.

## Figures and Tables

**Figure 1 ijerph-19-10287-f001:**
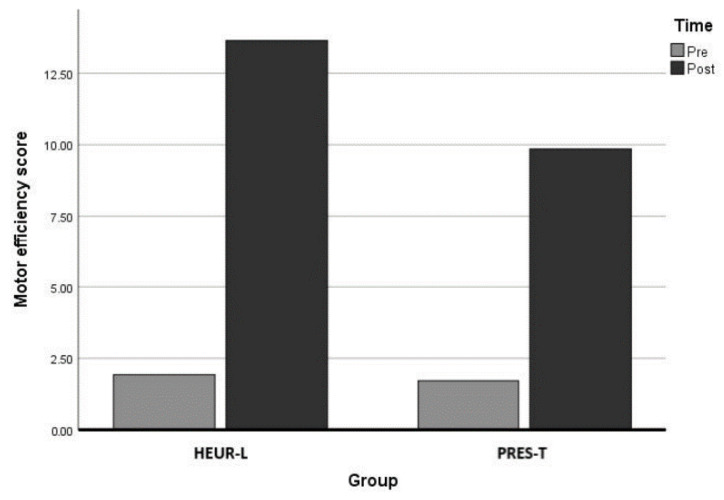
Differences in motor efficiency score according to group and time.

**Table 1 ijerph-19-10287-t001:** Training-protocol example of two groups using two different teaching methods.

HEUR-L: Ecological–Dynamic Approach	PRES-T: Cognitive Approach
Aim: To solicit/stimulate the emergence of new and spontaneous solutions to motor problems, exploiting executive variability.	Aim: To stabilise and perfect the motor programme by reducing executive variability.
Instructor’s role: The instructor did not demonstrate how to perform the circuit. He asked children to solve a motor problem through divergent discovery, guidance, feedback, etc., without giving the solution. To this end, he resorts to the use of educational practices, including altering the environment, game rules and techniques borrowed from psychology, including brainstorming at the beginning of the lesson, cooperative learning in the middle phase and circle time in the final phase. Each lesson was preceded by a 10-min brainstorming session on the focus of the day, followed by 5 min of circle time, in which each child expressed their own feelings.	Instructor’s role: Explain and demonstrate to the children how to perform the exercises to the best of their abilities through partial, varied and randomised practice, as well as correction of errors.
The circuits were the same. One of them consisted of: Step 1: running blindfolded Step 2: jumping on one leg Step 3: crawling while keeping upper limbs locked Step 4: overcoming an obstacle Step 5: walking backwards Another circuit consisted of: Step 1: throwing 3 balls of different sizes in a circle Step 2: crawling inside a tube Step 3: frog jumps inside circle Step 4: slalom-running around pins Step 5: kicking a ball between two pins	
The children were divided into mini-teams. Each member had to perform the proposed circuits, and a point was awarded for each exercise performed well. The time available was 15 min per circuit. The team with the most points within the time limit won. The various mini-circuits, consisting of motor patterns, music and mini-games, were preceded by an initial warm-up of 10 min and a final defatigue with static stretching and relaxation exercises lasting 10 min.	

**Table 2 ijerph-19-10287-t002:** Summary of two-way ANOVA results for the differences in motor efficiency according to group (HEUR-L and PRES-T), time (pre–post), and interaction groups × time.

Variables	Group	M ± SD		Variation			Effect Size (Partial η^2^)
		Pre	Post	Group	Time	Interaction	
Motor efficiency	HEUR-L	1.92 ± 2.30	13.64 ± 4.60	0.05 *	0.000 *	0.001 *	0.948
	PRES-T	1,71 ± 2.33	9.85 ± 1.51				0.369

* Significant differences at *p* < 0.05.

**Table 3 ijerph-19-10287-t003:** Differences in perceptions of the activity performed between HEUR-L and PRES-T.

		Subjects		Chi-Square Analysis	
Question	Options	HEUR-L	PRES-T	χ^2^	*p*
(1) Did you enjoy performing these exercises?	Yes	14	14		***
	No	0	0		
(2) Do you feel more improved in making friends with your peers or doing the exercises?	Friends	11	2	11.63	0.002
	Exercises	3	12		
(3) Do you think that performing exercises with certain difficulties can help you become better?	Yes	14	9	6.08	0.014
	No	0	5		

* Variable is constant.

**Table 4 ijerph-19-10287-t004:** Differences in awareness of the activity performed between HEUR-L and PRES-T.

		Subjects		Chi-Square Analysis	
Question	Options	HEUR-L	PRES-T	χ^2^	*p*
(1) Given the significant improvements in both groups, do you agree that these two types of intervention are effective and motivating in order to promote well-being and break down sedentary lifestyles?	Yes	14	14		***
	No	0	0		
(2) Given the greater improvements in HEUR-L than in PRES-T, do you agree that the educational practices are more optimal for developing/consolidating basic motor patterns?	Yes	13	1	6.46	0.143
	No	12	2		

* Variable is constant.

## Data Availability

Not applicable.

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
