# Peer review of "Effect of Teaching Methods on Motor Efficiency, Perceptions and Awareness in Children"

_ijerph, 2022, doi:10.3390/ijerph191610287_

Round 1

Reviewer 1 Report

The MS presents a study assessing the efficacy of two different teaching methods to favor motor efficiency, self-perceptions and awareness in a group of children. The topic is interesting and the MS is overall well-written. Hence, I suggest publication pending the revisions listed below.

Major

11. They found that both teaching methods were effective, maybe due to some underlying factors such as instructor enthusiasm and supportive style. Hence, I suggest adding in the Discussion some reflections on these possibilities. Suggested references

Hancox, J. E., Quested, E., Ntoumanis, N., & Thøgersen-Ntoumani, C. (2018). Putting self-determination theory into practice: Application of adaptive motivational principles in the exercise domain. Qualitative Research in Sport, Exercise and Health, 10(1), 75-91.

Keller, M. M., Hoy, A. W., Goetz, T., & Frenzel, A. C. (2016). Teacher enthusiasm: Reviewing and redefining a complex construct. Educational Psychology Review, 28(4), 743-769.

Moè, A. (2016). Does displayed enthusiasm favour recall, intrinsic motivation and time estimation? Cognition and Emotion, 30(7), 1361-1369.

Moè, A., & Katz, I. (2020). Self-compassionate teachers are more autonomy supportive and structuring whereas self-derogating teachers are more controlling and chaotic: The mediating role of need satisfaction and burnout. Teaching and Teacher Education, 96, 103173.

Reynders, B., Vansteenkiste, M., Van Puyenbroeck, S., Aelterman, N., De Backer, M., Delrue, J., ... & Broek, G. V. (2019). Coaching the coach: Intervention effects on need-supportive coaching behavior and athlete motivation and engagement. Psychology of Sport and Exercise, 43, 288-300.

Teixeira, D. S., Silva, M. N., & Palmeira, A. L. (2018). How does frustration make you feel? A motivational analysis in exercise context. Motivation and Emotion, 42(3), 419-428.

2. In the Introduction they stress that ‘motivating’ is the way to favor the process. Nevertheless, motivation was not directly trained. I would suggest including this issue in the Discussion/Future Avenues: future research should prompt not only strategies (as depicted in Table 1), but also some motivational aspects, such as effort attribution, and an incremental view of motor performances as well as intrinsic motivation (maybe via perceived need satisfaction). Suggested references

Moè, A. (2016). Teaching motivation and strategies to improve mental rotation abilities. Intelligence, 59, 16-23.

Orvidas, K., Burnette, J. L., & Russell, V. M. (2018). Mindsets applied to fitness: Growth beliefs predict exercise efficacy, value and frequency. Psychology of Sport and Exercise, 36, 156-161.

Roberts, G. C., & Nerstad, C. G. (2020). Motivation: achievement goal theory in sport and physical activity. In The Routledge International Encyclopedia of Sport and Exercise Psychology (pp. 322-341). Routledge.

 3. As future avenues/limitations, I would add assessing the instructor perceptions of enjoyment, preferences, perceived effort….of the two teaching methods and the method habitually adopted. Future research should also assess the long term effects, in the follow-up and include larger samples.

Minor

1Instead of writing ‘group 1’ or ‘group 2’ please choose a label since the beginning and use it throughout. Accordingly, please replace in Graph 1, 1 and 2 with the label of the teaching method

2I would structure the Introduction and the Discussion with sub-heading for ease reading

Author Response

Dear Reviewer 1,

thank you for reviewing our manuscript. We have taken up almost all suggestions, highlighting them in yellow in the full text. We have also responded to some critical issues raised, as follows.

REVIEWER 1 (R1) / AUTHORS (A)

R1: The MS presents a study assessing the efficacy of two different teaching methods to favor motor efficiency, self-perceptions and awareness in a group of children. The topic is interesting and the MS is overall well-written. Hence, I suggest publication pending the revisions listed below. They found that both teaching methods were effective, maybe due to some underlying factors such as instructor enthusiasm and supportive style. Hence, I suggest adding in the Discussion some reflections on these possibilities. Suggested references

  • Hancox, J. E., Quested, E., Ntoumanis, N., & Thøgersen-Ntoumani, C. (2018). Putting self-determination theory into practice: Application of adaptive motivational principles in the exercise domain. Qualitative Research in Sport, Exercise and Health, 10(1), 75-91.
  • Keller, M. M., Hoy, A. W., Goetz, T., & Frenzel, A. C. (2016). Teacher enthusiasm: Reviewing and redefining a complex construct. Educational Psychology Review, 28(4), 743-769.
  • Moè, A. (2016). Does displayed enthusiasm favour recall, intrinsic motivation and time estimation? Cognition and Emotion, 30(7), 1361-1369.
  • Moè, A., & Katz, I. (2020). Self-compassionate teachers are more autonomy supportive and structuring whereas self-derogating teachers are more controlling and chaotic: The mediating role of need satisfaction and burnout. Teaching and Teacher Education, 96, 103173.
  • Reynders, B., Vansteenkiste, M., Van Puyenbroeck, S., Aelterman, N., De Backer, M., Delrue, J., ... & Broek, G. V. (2019). Coaching the coach: Intervention effects on need-supportive coaching behavior and athlete motivation and engagement. Psychology of Sport and Exercise, 43, 288-300.
  • Teixeira, D. S., Silva, M. N., & Palmeira, A. L. (2018). How does frustration make you feel? A motivational analysis in exercise context. Motivation and Emotion, 42(3), 419-428.

A: Thanks for the suggestions. We added it in the discussion using these references.

R1: In the Introduction they stress that ‘motivating’ is the way to favor the process. Nevertheless, motivation was not directly trained. I would suggest including this issue in the Discussion/Future Avenues: future research should prompt not only strategies (as depicted in Table 1), but also some motivational aspects, such as effort attribution, and an incremental view of motor performances as well as intrinsic motivation (maybe via perceived need satisfaction). Suggested references

  • Moè, A. (2016). Teaching motivation and strategies to improve mental rotation abilities. Intelligence, 59, 16-23.
  • Orvidas, K., Burnette, J. L., & Russell, V. M. (2018). Mindsets applied to fitness: Growth beliefs predict exercise efficacy, value and frequency. Psychology of Sport and Exercise, 36, 156-161.
  • Roberts, G. C., & Nerstad, C. G. (2020). Motivation: achievement goal theory in sport and physical activity. In The Routledge International Encyclopedia of Sport and Exercise Psychology (pp. 322-341). Routledge.

 A: Thanks for the suggestions. We added it in the discussion with some of these references.

R1: As future avenues/limitations, I would add assessing the instructor perceptions of enjoyment, preferences, perceived effort….of the two teaching methods and the method habitually adopted. Future research should also assess the long term effects, in the follow-up and include larger samples.

A: Thanks for the suggestions.We done it.

R1: Instead of writing ‘group 1’ or ‘group 2’ please choose a label since the beginning and use it throughout. Accordingly, please replace in Graph 1, 1 and 2 with the label of the teaching method

A: Thanks for the suggestions. We done it.

R1: I would structure the Introduction and the Discussion with sub-heading for ease reading

A: Thanks for the suggestions. We done it.

Thank you for your time. We appreciate it. If you have any questions do not hesitate to ask us.

Reviewer 2 Report

The summary does not reflect the content. Very cursory literature analysis and very old items. Selection of research methodology unexplained, lack of information about the research gap is not the purpose of this research. Why such a group? where does the split come from? Why children at this age? No explanation of the teaching methods used. They are different? why were these chosen? No limitations to the study of their use and further possibilities.

Author Response

Dear Reviewer 2,

thank you for reviewing our manuscript. We have taken up almost all suggestions, highlighting them in yellow in the full text. We have also responded to some critical issues raised, as follows.

REVIEWER 2 (R2) / AUTHORS (A)

R2: The summary does not reflect the content. Very cursory literature analysis and very old items.

A: Thanks for the suggestions. We have rewritten the abstract and the introduction more clearly and divided this last one into paragraphs, updating some references.

R2: Selection of research methodology unexplained, lack of information about the research gap is not the purpose of this research.

A: Thanks for the suggestions. We explained the methods section more clearly. The problem is the prevalence of poor PA in children, resulting in poor ability to perform basic motor patterns. It is important to find some strategies that the instructor can adopt to improve awareness of the importance of PA for health and wellness, over motor efficiency, through the two teaching methods, previously addressed. Consequently, the aim was to compare the effects of prescriptive teaching and heuristic learning, to verify the most suitable to develop motor efficiency; in addition, to verify the level of enjoyment and self-efficacy through questions on perception and, subsequently, awareness of the activity performed distinctly from perception (line 101- 112).

R2: Why such a group? where does the split come from? Why children at this age?

A: Sampling is of convenience, meaning that children involved in extracurricular activities were purposely chosen for the study because they are close to the researcher (and this is also one of the limitations of the study). Children from the age of 5 years were chosen, as it is believed that from the beginning of primary school children begin to have a greater awareness of the activities they perform, as the aim of the study is also to make children aware of the importance of practising physical activity consistently in order to counteract sedentariness and physical inactivity. They were randomly divided into two groups in order to propose a heuristic methodology to one group and a prescriptive one to the other (Line 115-126).

R2: No explanation of the teaching methods used. They are different? why were these chosen?

A: We have added this information from line 139 to 161. Prescriptive teaching is instructor-centred, whereas heuristic learning is student-centred. Thanks for the suggestion.

R2: No limitations to the study of their use and further possibilities.

A: You can find the limitation of the study in the discussion (line 317 – 325).

Thank you for your time. We appreciate it. If you have any questions do not hesitate to ask us.

Round 2

Reviewer 2 Report

The authors complied with the comments and introduced changes. Thanks to them, the article is much more understandable.